# Role of aspirin on colorectal cancer risk and bacterial translocation to bloodstream

Silvia Mignozzi[1]*, Giuseppe De Pinto[1], Simone Guglielmetti[2], Patrizia Riso[3], Marcello Cintolo[4], Roberto Penagini[5,6], Giorgio Gargari[3], Mirko Marino[3], Clorinda Ciafardini[6], Monica Ferraroni[1,7], Rossella Bonzi[1], Massimiliano Mutignani[4], Carlo La Vecchia[1], Marta Rossi[1]

1 Department of Clinical Science and Community Health, *Dipartimento di Eccellenza 2023-2027*, University of Milan, Milan, Italy, 2 Department of Biotechnology and Biosciences, University of Milan Bicocca, Milan, Italy, 3 Department of Food, Environmental and Nutritional Sciences, University of Milan, Milan, Italy, 4 Digestive and Interventional Endoscopy Unit, ASST Grande Ospedale Metropolitano Niguarda, Milan, Italy, 5 Department of Pathophysiology and Transplantation, University of Milan, Milan, Italy, 6 Gastroenterology and Endoscopy Unit, Fondazione IRCCS Ca' Granda Ospedale Maggiore Policlinico, Milan, Italy, 7 Fondazione IRCCS Ca' Granda Ospedale Maggiore Policlinico, Milan, Italy

* silvia.mignozzi@unimi.it

## Abstract

An involvement of gut microbiota on the possible role of aspirin against intestinal adenoma (IA) and colorectal cancer (CRC) has been suggested. To further investigate this issue, we analyzed data from an Italian case-control study including 100 incident histologically confirmed CRC cases, as well as 100 IA and 100 controls without lesions from colonoscopy, matched to cases by center, sex and age. Serum zonulin was assessed by ELISA kit and blood bacterial DNA by qPCR and 16S rRNA gene profiling. Fifty-eight subjects (19.3%) reported aspirin use of ≥ 100 mg/day for cardiovascular prevention for at least six months. To evaluate the relationship between aspirin and IA and CRC risks, the odds ratios (OR) of IA and CRC and the corresponding 95% confidence intervals (CI) for aspirin use were estimated using a logistic regression model conditioned on the matching variable and adjusted for education and a model adjusted for several potential confounders including BMI and cardiovascular diseases. We evaluated whether the levels of zonulin and bacterial DNA data were different in aspirin users *vs* non-users through the rank sum and chi-square tests. Aspirin use was associated with a reduced risk of IA (OR = 0.45, 95% CI = 0.21-0.94) and CRC (OR = 0.43, 95% CI = 0.19-0.96). Similar results were obtained using the fully adjusted model. We found lower genera and operational taxonomic units (OTUs) richness of blood bacterial community in aspirin users *vs* non-users overall and in cases and controls. The genera *Cutibacterium*, *Sphingomonas*, *Gaiella*, *Delftia* and *Romboutsia,* order *Microtrichales* and class *Deltaproteobacteria* were different according to aspirin use. This study provides additional data on the favorable role of aspirin on IA and CRC risks and supports the hypothesis of an involvement of intestinal bacterial translocation to the bloodstream.

**Data availability statement:** All the reads are publicly available in the European Nucleotide Archive (ENA) with the accession number: PRJEB46474. All data are in the paper and/or Supporting Information files."

**Funding:** "Data collection was supported by the Italian Foundation for Cancer Research (AIRC) (My First AIRC grant No. 17070) (MR). Data analysis was supported by the grant PRIN 2022 PNRR (no. P20229A9S5) from the Italian Ministry of University and Research (MUR). The funders had no role in study design, data collection and analysis, decision to publish, or preparation of the manuscript. There was no additional external funding received for this study."

**Competing interests:** The authors have declared that no competing interests exist.

## Introduction

Inverse associations between aspirin use and intestinal adenoma (IA) and colorectal cancer (CRC) risk have been reported [1–3]. Biomolecular pathways explaining the favorable role of aspirin on CRC are not completely clear, but the role of COX-1 and COX-2 is well-recognized [4,5]. COX-2 inhibition leads to a drop in prostaglandin E2, that affects apoptosis and cell proliferation [5]. In addition, aspirin as well as other nonsteroidal anti-inflammatory drugs alter the gut microbiome, partly explaining their anti-carcinogenic effect on colorectal carcinogenesis [6]. This can promote macrophages production, which has a relevant role in the surveillance of the intestinal microbiota, resulting in a higher release of several proinflammatory mediators [6,7]. Furthermore, at tissue level, the deacetylation of aspirin in the liver and stomach causes the formation of salicylic acid, which permeates human gastrointestinal tissues, potentially influencing the intestinal bacterial taxa [8,9]. Thus, the mechanism explaining the association between aspirin use and CRC risk can be related to an impact of aspirin use on gut microbiota, via a systemic or local mechanism or both. Variations in the intestinal microbiota have been linked to increased intestinal permeability and bacterial translocation in CRC patients [10–12]. In addition, circulating levels of zonulin - the protein responsible for regulating tight junctions and crucial for protecting the intestinal barrier against bacterial translocation -and blood bacterial 16S rRNA gene copies - a measure of blood bacterial load - have been positively associated with CRC risk [13–16].

The aim of this study is to assess the associations between aspirin use and IA and CRC risks and to explore whether the intestinal bacterial translocation to the bloodstream (estimated through serum zonulin and blood 16S rRNA gene) is involved in the mechanisms behind these associations, using data from an Italian case-control study [14].

## Materials and methods

We conducted a case-control study comprising 100 incident histologically confirmed CRC cases, 100 subjects with IA, and 100 healthy controls (without CRC and IA) in Milan, Italy, between 1 May 2017 and 1 November 2019 [14].

Subjects were recruited in two general hospitals by trained health care professionals among eligible outpatients or inpatients scheduled for a colonoscopy, aged 20-85 years old (S1 Fig in S1 File). Two pathologists determined CRC and IA cases and controls by revising colonoscopy and histological examinations. One CRC case was matched to one IA and one healthy control by study center, sex, and age (±5). Chronic bowel inflammatory diseases, liver failure, high-grade kidney and heart failure, immunodeficiency, previous cancer, hospitalization in the last month for immune, autoimmune and inflammatory diseases were exclusion criteria [14]. Approximately 25% did not satisfy the eligibility criteria and less than 2% of eligible subjects refused to participate. Informed consent was obtained from all participants in the study. The protocol was approved by the ethical committees of the involved hospitals: Fondazione IRCCS Ca' Granda Ospedale Maggiore Policlinico (No. 742-2017) and ASST Grande Ospedale Metropolitano Niguarda (No. 477-112016).

Trained and blinded interviewers conducted a face-to-face interview with each subject before colonoscopy. The questionnaire included information on socio-demographics, anthropometric measures, lifestyle habits (including occupational physical activity), supplement and antibiotic use and medical history. A section contained information on the use of selected drugs at least once a week for more than 6 months, with data on start date, end date, weekly frequency and therapeutic indication. From this section, we extracted information to identify a long-term aspirin user as a subject who reported aspirin use at ≥ 100 mg/day for at least 6 months for cardiovascular prevention. Dietary information was collected

through a food frequency questionnaire including 78 food items. From this data, total energy intake was also estimated for each subject on the basis of an Italian food composition database [17].

Blood samples were collected in 7 mL EDTA tube and in 3 mL blank tube before the colonoscopy. Part of the EDTA sample was immediately stored at −80 °C for the microbiomic analysis. The sample without anticoagulant was processed and centrifuged (1400 g × 15 min, 4 °C) and the serum stored at −80 °C for circulating zonulin analysis [13,14].

DNA extraction, quantitative polymerase chain reaction (qPCR) experiments and sequencing of 16S rRNA gene amplicons were performed by Vaiomer SAS, Labège, France [14]. An aliquot of 0.25 ml of whole blood was used for DNA extraction. Real-time qPCR amplification was performed using panbacterial primers EUBF50-TCCTACGGGAGGCAGCAGT-30 and EUBR 50-GGACTACCAGGGTATCTAATCCTGTT-30. The number of bacterial 16S rRNA gene copies per µl of blood was measured as the abundance of 16S rRNA by qPCR in triplicate normalized using a plasmid-based standard range. MiSeq Illumina technology was applied to compute 16S rRNA gene taxonomic profiling. Sequences were analyzed using Vaiomer bioinformatic pipeline, that includes the FROGS v1.4.0 to identify operational taxonomic units (OTUs) and the Blast + v2.2.30 to perform the taxonomic assignment from *Phylum* to *Genera*, against the Silva 132 Parc database. Four samples (only one among aspirin users) were excluded from taxonomic profiling analysis because they did not reach the threshold of needed reads. We made all the reads publicly available in the European Nucleotide Archive (ENA) with the accession number: PRJEB46474 (uploaded on 6th September 2021 and available indefinitely). The data were used for this research purpose from September 2022 to September 2023. Observed-features, Chao1, Shannon, Simpson and InvSimpson alpha-diversity indices were computed using R PhyloSeq v1.14.0 package [14].

Serum zonulin levels were quantified using the ELISA kit (cat. # K5601) from Immunodiagnostik® (Bensheim, Germany) [13]. Serum samples were defrosted at room temperature and applied to a 96-well plate precoated with polyclonal anti-zonulin antibodies. A biotinylated zonulin tracer was added to each sample. Peroxidase-labelled streptavidin was added to bind the biotinylated tracer. After the reaction fluorescence was measured at 450 nm using a Tecan Infinite F200 plate reader. Zonulin levels were quantified by constructing a standard curve using a 4-parameter algorithm [13].

## Statistical analyses

We computed the odds ratios (OR) of IA and CRC and the corresponding 95% confidence intervals (CI) for aspirin use compared to non-use through a logistic regression model conditioned on the matching variable and adjusted for education (<7, 7-11, ≥ 12 years). We also considered models further adjusted for body mass index (BMI), physical activity, cardiovascular diseases, diabetes, antibiotic use and total energy intake.

We compared the distributions of serum zonulin and blood bacterial 16S rRNA gene copies and alpha diversity indices (Observed-features, Chao1, Shannon, Simpson and InvSimpson) according to aspirin use, overall and among controls, IA and CRC subjects, carrying out two-tailed Wilcoxon rank sum test.

For bacteria with at least a representation of about 5%, we compared the distributions of relative abundances of OTUs and taxa at each taxonomical level between aspirin users and non-users overall, using the Wilcoxon rank sum test. We also evaluated the difference of prevalence of OTUs and taxa using a chi-square test and applying a crude logistic regression model and a model adjusted for a term of control/IA/CRC. When we found a series of taxa associated with aspirin use belonging to the same phylogenetic branch, we selected the taxa at minimum level of the branch and we showed their distributions through boxplot.

## Results

The distribution of aspirin users and non-users by study center, sex, age and education is given in Table 1. Overall, aspirin users were 58 (19.3%) and non-users 242 (80.7%). Aspirin users were more frequently men (70.7%), older (77.2%: mean age was 72.5 for aspirin users and 64.4 for non-users) and with a lower degree of education (55.2%).

Aspirin use was associated with a reduced risk of IA (OR = 0.45, 95% CI = 0.21-0.94) and of CRC (OR = 0.43, 95% CI = 0.19-0.96) as compared to non-use (Table 2). The OR of CRC was 0.38 (95% CI = 0.16-0.91) and the OR of IA was 0.41 (95% CI = 0.18-0.94) from the fully adjusted model.

Aspirin users had higher levels of serum zonulin (median = 30.7 ng/mL) as compared to non-users (median = 28.5 ng/mL) among controls (p = 0.02) (Table 3). Aspirin users had lower levels of serum zonulin (median = 27.2) as compared to non-users (median = 29.1) among IA cases, without a significant difference (p = 0.1). No difference was found in terms of zonulin, in CRC cases and overall. With reference to 16S rRNA gene analysis, higher copies were found in aspirin users (median = 7786.2) *versus* non-users (median = 7084.5) overall, without

**Table 1. Distribution of aspirin users and non-users by study center, sex, age and years of education. Italy, 2017-2019.**

|  | Aspirin use | Non aspirin use | X² p-value |
|---|---|---|---|
|  | n = 58 | n = 242 |  |
|  | n. (%) | n. (%) |  |
| **Study center** |  |  |  |
| Niguarda | 35 (60.3) | 160 (66.1) |  |
| Policlinico | 23 (39.7) | 82 (33.9) | 0.4 |
| **Sex** |  |  |  |
| Male | 41 (70.7) | 145 (59.9) |  |
| Female | 17 (29.3) | 97 (40.1) | 0.13 |
| **Age group** |  |  |  |
| <70 | 19 (32.8) | 155 (64.1) |  |
| 70-74 | 14 (24.1) | 34 (14.1) |  |
| ≥75 | 25 (43.1) | 53 (21.9) | <0.0001 |
| Mean (SD) | 72.5 (8.0) | 64.4 (11.6) | <0.0001[b] |
| **Education (years)[a]** |  |  |  |
| <12 | 32 (55.2) | 99 (40.9) |  |
| ≥12 | 26 (44.8) | 142 (58.9) | 0.05 |

[a]The sum does not add up to the total because of one missing value.

[b]p computed using a t-test.

**Table 2. Distribution of aspirin users and non-users by controls, intestinal adenoma (IA) and colorectal cancer (CRC) patients and the odds ratios (OR) and the corresponding 95% confidence intervals (CI) of IA and CRC patients for aspirin use. Italy 2017-2019.**

|  | Aspirin use<br>n. (%) | Non-aspirin use<br>n. (%) | OR[a] (CI) |
|---|---|---|---|
| Controls | 27 (46.6) | 73 (30.2) |  |
| IA | 16 (27.6) | 84 (34.7) | 0.45 (0.21-0.94) |
| CRC | 15 (25.9) | 85 (35.2) | 0.43 (0.19-0.96) |

[a]Estimated from conditioned logistic regression models adjusted for education.

a significant difference (p = 0.1). Similarly, a higher number of copies were found in aspirin users (median = 7794.8 copies per μL) *versus* non-users (median = 6970.2 copies per μL) among controls (p = 0.07). We found a lower bacterial richness in aspirin users compared to non-users, in genera in terms of both Observed-features (median = 27.0 *vs.* 29.0, p = 0.02) and Chao1 (median = 37.0 *vs.* 45.0, p = 0.004) indices and in OTUs in terms of both Observed-features (median = 34.0 vs. 36.0, p = 0.02) and Chao1 (median = 54.5 *vs.* 66.0, p = 0.04) indices overall. Similar results were found among CRC cases and controls. No difference between aspirin users and non-users was found in terms of Shannon, Simpson and InvSimpson alpha-diversity indices.

Relative abundance and/or presence of the genera *Cutibacterium*, *Sphingomonas*, *Gaiella*, *Delftia* and *Romboutsia*, the order *Microtrichales* and the class *Deltaproteobacteria* were significantly different according to aspirin use (p ≤ 0.05) (S1 Table in S1 File). In addition, sequencing reads assigned to *Mitochondria* were absent in aspirin users and present in 21 non-users.

Fig 1a–g show the relative abundance and prevalence of selected taxa in aspirin users and non-users with the corresponding p of test for heterogeneity. For the genera *Cutibacterium* (Fig 1a) and *Sphingomonas* (Fig 1b), there were no significant differences in terms of presence between aspirin users and non-users. However, a lower relative abundance of these genera occurred in aspirin users (p = 0.009 and p = 0.01, respectively). In addition, aspirin users had an increased prevalence of *Gaiella* (Fig 1c) and *Delftia* (Fig 1d) genera (χ-square p = 0.02; OR = 3.49, 95% CI = 1.15-10.54 and χ-square p = 0.03; OR = 2.07, 95%CI = 1.06-4.03, respectively) and higher relative abundances (p = 0.02 and p = 0.03, respectively). For genus *Romboustia* (Fig 1e), there was a significant reduction in terms of prevalence in aspirin users as compared to non-users (χ-square p = 0.04; OR = 0.25, 95% CI = 0.06-1.05) and a lower relative abundance (p = 0.04). For *Microtrichales* order (Fig 1f), aspirin users had an increased prevalence (χ-square p = 0.04; OR = 2.49, 95% CI = 1.04-5.93) and a higher relative abundance

**Table 3. Distributions of serum zonulin, blood bacterial 16S rRNA gene copies and Observed-features and Chao1 alpha-diversity indices for genera and OTUs according to aspirin use overall and among controls, intestinal adenoma (IA) and colorectal cancer (CRC) subjects. Italy 2017–2019.**

| | Zonulin | | | Bacterial 16S rRNA gene data | | | | | | | | | | | | | | | | |
| | | | | 16S rRNA gene copies | | | Observed genera index | | | Chao1 genera index | | | Observed OTUs index | | | Chao1 OTUs index | | |
| | Median (I-III IQ) | | p[a] | Median (I-III IQ) | | p[a] | Median (I-III IQ) | | p[a] | Median (I-III IQ) | | p[a] | Median (I-III IQ) | | p[a] | Median (I-III IQ) | | p[a] |
| | Aspirin use | Non aspirin use | | Aspirin use | Non aspirin use | | Aspirin use | Non aspirin use | | Aspirin use | Non aspirin use | | Aspirin use | Non aspirin use | | Aspirin use | Non aspirin use | |
| Overall | 29.5 (27.1-32.6) | 29.6 (26.8-32.5) | 0.7 | 7786.2 (5968.9-10290.6) | 7084.5 (5599.3-9241.9) | 0.1 | 27.0 (24.0-31.0) | 29.0 (25.0-33.0) | 0.02 | 37.0 (31.0-50.3) | 45.0 (35.5-57.5) | 0.004 | 34.0 (30.0-38.0) | 36.0 (32.0-44.0) | 0.02 | 54.5 (42.3-73.0) | 66.0 (50.0-87.5) | 0.04 |
| Controls | 30.7 (28.3-33.0) | 28.5 (26.1-30.8) | 0.02 | 7794.8 (6047.3-10290.6) | 6970.2 (5633.7-8492.6) | 0.07 | 26.5 (24.0-30.0) | 29.0 (25.0-33.5) | 0.1 | 36.5 (29.5-49.0) | 46.1 (36.0-61.0) | 0.02 | 33.0 (30.0-34.0) | 35.0 (31.5-44.0) | 0.05 | 51.3 (44.5-66.0) | 69.3 (49.0-87.4) | 0.007 |
| IA | 27.2 (25.9-29.2) | 29.1 (26.9-32.1) | 0.1 | 7167.32 (5830.1-9276.7) | 7095.8 (5535.7-9187.3) | 0.7 | 28.0 (24.0-35.5) | 28.0 (25.0-32.0) | 0.9 | 49.3 (34.3-57.2) | 43.5 (34.0-55.5) | 0.5 | 37.0 (30.5-44.5) | 35.0 (31.0-40.0) | 0.7 | 82.5 (54.8-130.5) | 64.9 (49.0-86.5) | 0.1 |
| CRC | 30.5 (28.8-32.6) | 30.9 (27.2-33.6) | 0.8 | 8047.4 (5085.0-11947.1) | 7435.2 (9187.3-9772.8) | 0.5 | 27.0 (22.0-31.0) | 30.0 (26.0-34.0) | 0.05 | 34.0 (28.0-44.0) | 44.8 (38.0-46.5) | 0.001 | 34.0 (27.0-39.0) | 37.0 (33.0-45.0) | 0.08 | 52.0 (39.3-70.0) | 63.0 (51.0-88.0) | 0.04 |

[a]p for heterogeneity estimated from the Wilcoxon rank-sum.

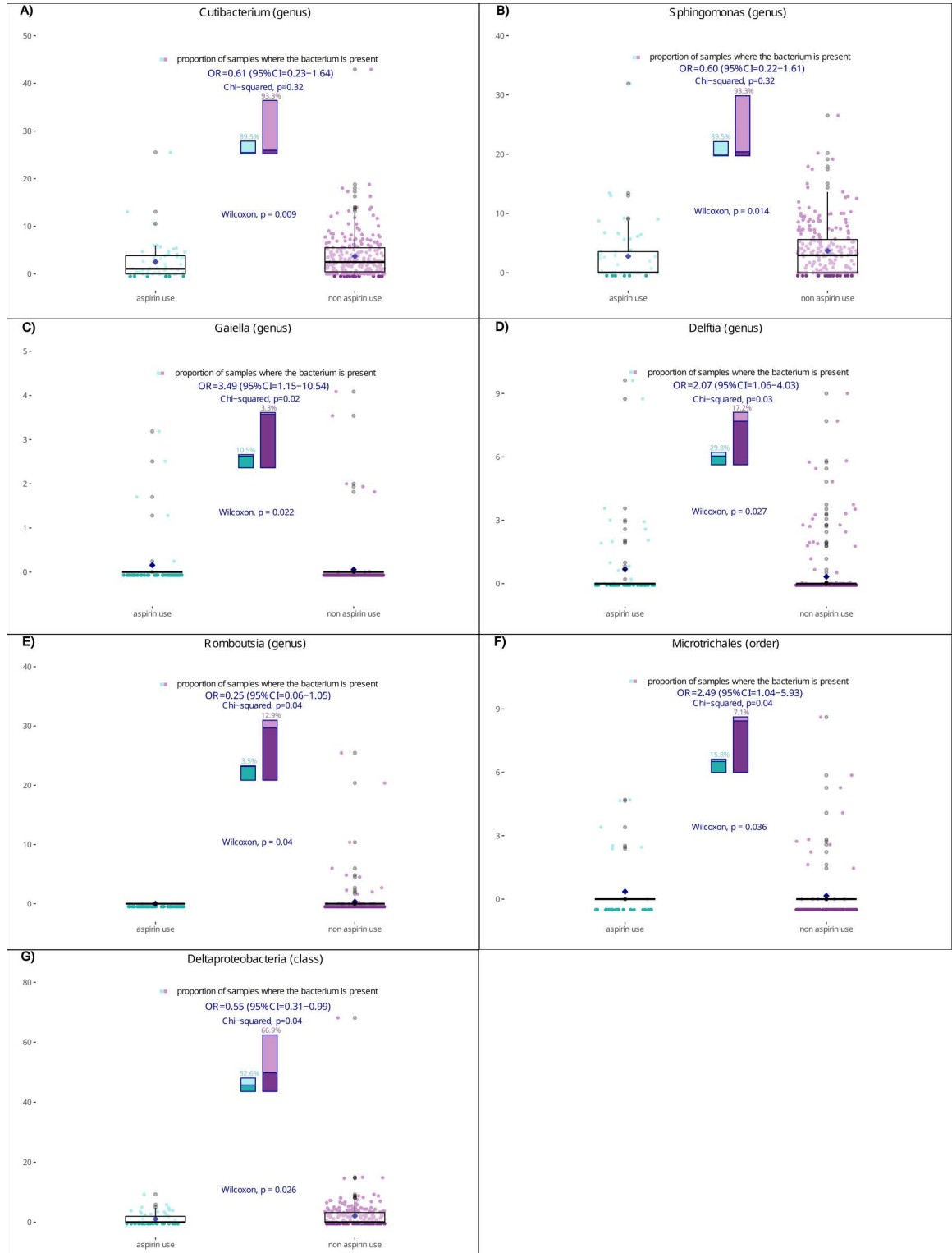

**Fig 1.  a-g. Relative abundance and prevalence of the genera Cutibacterium (a), Sphingomonas (b), Gaiella (c), Delftia (d) and Rombustia (e), the order Microtrichales (f) and the class Deltaproteobacteria (g) according to aspirin use and corresponding p for test.** Italy 2017-2019. The odds ratios (OR) and the corresponding 95% confidence intervals (CI) were estimated from logistic regression models adjusted for a term of control/intestinal adenoma/colorectal cancer.

(p = 0.04). Finally, aspirin users had a reduced prevalence of *Deltaproteobacteria* (Fig 1g) (χ-square p = 0.04; OR = 0.55, 95% CI = 0.31-0.99) and a lower relative abundance (p = 0.03).

## Discussion

Our study found a reduced risk of IA and of CRC for aspirin users compared to non-users. Blood bacterial community showed a lower richness among aspirin users. The genera *Cutibacterium*, *Sphingomonas*, *Gaiella*, *Delftia* and *Romboutsia*, the order *Microtrichales* and the class *Deltaproteobacteria* were differently distributed according to aspirin use.

A meta-analysis reported an inverse association between regular aspirin use and the risk of CRC [1]; the pooled relative risk was 0.62 (95% CI = 0.56-0.69) in case-control studies, 0.77 (95% CI = 0.72-0.82) in cohort studies and 0.73 (95% CI 0.69-0.78) overall, with a linear dose-risk relation. In our data, the OR of CRC for aspirin use appeared to be lower (OR = 0.43, 95% CI = 0.19-0.96). This can be due to some residual confounding or to the limited sample size. More recently, a pooled analysis of two large US cohort studies including 94,540 participants and 1431 incident cases, aged 70 years or older [18], found that regular aspirin use was associated with a lower risk of CRC with a relative risk of 0.80 (95% CI = 0.72-0.90). Starting regular aspirin use after 70 years was not associated with lower CRC risk, whereas starting aspirin use before the age of 65 led to a reduced risk of CRC. In our study population, 75.9% (44 subjects) of aspirin users started regular aspirin use at the age of 70 at the latest and only 20.7% (12 subjects) after the age of 70.

In our previous investigations, we showed that serum zonulin and blood bacterial 16S rRNA gene levels (circulating biomarkers of intestinal permeability and bacterial translocation) were higher in CRC cases, likely due to the tumor itself or to a cancer-associated inflammatory condition [13,14]. If aspirin acts against CRC development through mechanisms linked to bacterial translocation, we would have expected aspirin users to have lower levels of these biomarkers. However, in our data, no significant differences in terms of zonulin and 16S rRNA gene copies were found overall, whereas higher levels were observed in aspirin users as compared to non-users among controls. This can be due to an aspirin biochemical action [19] (more evident in controls than in CRC affected patients) or to a high prevalence of inflammatory conditions, which have been associated with higher zonulin levels [20] in our controls. In the latter case, zonulin and 16S rRNA gene copies can have acted as confounding factors or effect modifiers on the relationship between aspirin use and CRC risk. In order to elucidate this aspect, we performed further analyses by adding terms for zonulin and 16S rRNA gene copies in our core model, and found that the OR of CRC for aspirin use was lower (OR = 0.18, 95% CI = 0.05-0.67) as compared to the one obtained from the core model (OR = 0.43, 95% CI = 0.19-0.96). Moreover, we computed the ORs of CRC for aspirin use across strata of high and low zonulin (<, ≥ 29.2 ng/mL = median among controls) and of high and low 16S rRNA gene copies (<, ≥ 7124.8 gene copies per μL = median among controls). The four ORs were similar to the adjusted OR, supporting the hypothesis that zonulin and 16S gene copies act as confounders, i.e., the association between aspirin use and CRC risk was altered by these biomarkers, unequally distributed throughout our exposure and outcome [21].

Microbiome in different tissues has been linked to several inflammatory diseases as well as CRC [9,22]. Oral drugs including aspirin are able to alter the gut microbiome [23]. In our study, we did not analyze the bacterial community structure at intestinal level but we taxonomically profiled the bacterial DNA isolated from peripheral blood. This analysis revealed a lower richness in aspirin users at genera and OTUs levels overall and in both cases and controls. Aspirin users can develop a tendency to down-regulate diversity in microbiota composition [23,24]. Through enterocyte lipoxin receptor, aspirin can decrease Tumor

Necrosis Factor α (TNF-α) and Interleukin-8 levels [25], and this can alter immune response to microbial invasion and translocation, also in terms of bacterial selectivity [26]. On the other hand, our result may be confounded by a higher presence in aspirin users of cardiovascular diseases, which have been associated with reduced richness [27–29]. However, our results did not change excluding subjects with cardiovascular diseases.

We also found differences in blood bacterial profiling between aspirin users and non-users. This is consistent with the hypothesis that the chemopreventive action of aspirin on CRC occurs through, or involves alterations of microbiota and taxa abundances in the gut. Among aspirin users, we found a lower relative abundance of the *Sphingomonas* genus in the blood, which may have a relationship, albeit unclear, with the development of colitis-associated cancer [30]. In fact, an Italian prospective study involving 7 patients with colitis-associated cancer, 10 patients with sporadic cancer and 10 controls, reported a higher relative abundance of *Sphingomonas* genus in the tumor mucosa-associated microbiota in colitis-associated cancer, compared to sporadic cancer [30]. Among aspirin users, we also found a lower relative abundance of the *Cutibacterium* genus. *Cutibacterium* is mostly present in the skin microbiota and is more frequently present in prostate cancer tissue than in normal prostate tissue [31–33]. A Swedish cohort study, including 137 men with prostate cancer, found a positive association between the presence of *Cutibacterium* and infiltration of regulatory T cells (Tregs) in the stroma and tumour epithelia [34]. Macrophages cultured with *Cutibacterium* in vitro increase the expression of immunosuppressive genes [35,36]. Moreover, prostate cancer patients with *Cutibacterium* had a higher infiltration of Tregs than their uninfected counterparts. This suggests that *Cutibacterium* may contribute to an immunosuppressive tumor environment critical for prostate cancer progression. Since Tregs are commonly found in solid tumors, including CRC, and promote immunosuppression through various mechanisms [35,36], the presence of *Cutibacterium* can be linked to the development of CRC. Aspirin users had increased prevalence and relative abundance of *Gaiella* genus. In a Chinese study involving tumor, peritumor and normal tissues extracted from 60 gastric cancer patients without preoperative chemotherapy, *Gaiella* was found to be inversely correlated with gastric Foxp3 + Treg, which are increased in tumoral and peritumoral tissues compared to normal ones [37]. The mechanism behind this association is unclear, but this suggests a similarity with CRC. Generally, members of the phylum *Actinomycetota* (formerly *Actinobacteria*), such as genera *Cutibacterium* and *Gaiella*, appear to play a role in cancer prevention and treatment due to their capacity to synthesize anti-cancer compounds [38]. Within the intestinal ecosystem, *Actinomycetota* facilitates the conversion of linoleic acid to conjugated linoleic acid, thus improving immune function. Furthermore, these bacteria contribute to improve the inflammatory response by increasing levels of Lipopolysaccharide and TNF [38]. We found that aspirin users had significantly increased prevalence and relative abundance of *Delftia* genus. A study from the Tianjin Union Medical Centre analyzed the intestinal mucosal microbiota of 37 patients with stage II CRC, finding higher relative abundances of *Delftia* genus as compared to 26 healthy patients [39], in contrast with our results. Among aspirin users, we found a lower relative abundance of *Romboustia* genus. In a Chinese study, involving 21 patients with primary hepatocellular carcinoma (HCC) and 21 healthy first-degree relatives of cases, fecal *Romboustia* was strongly inversely associated with HCC [40]. This bacterium was also found to be negatively correlated with alpha-fetoprotein, a cancer biomarker involved in gastrointestinal cancers [40].

In order to reduce information bias in our study, trained and blinded interviewers administered the questionnaire before endoscopy. This made participants unaware of cancer diagnosis, reducing overreporting of cases. With regard to selection bias, cases and controls were recruited from the same catchment areas by fulfilling the procedures of an *ad hoc* developed protocol. The colonoscopic examination for each participant in the recruitment phase

ensured the inclusion of CRC-free controls and patient with IA, obtaining a complete pathogenic sequence of CRC. With reference to confounding, we were able to allow for a number of relevant covariates, including study center, sex, age, education, BMI and cardiovascular diseases. The study can be underpowered since a sample of about 200 subjects, with a ratio of controls to cases of 1 and percentage of exposed controls of 30%, allows the assessment of an OR of 0.4, with 80% of statistical power and 0.05 of type I error [41]. Another limitation is that no biological material other than blood were available. This did not allow us to evaluate the origin of the microbial materials in blood and to deepen the biochemical mechanisms of aspirin in protecting from CRC. Moreover, no significant results were found when we applied the Benjamin-Hochberg correction to non-parametric tests on genera and OTUs.

This study provides additional data on the possible inverse association between aspirin use and CRC risk [42,43] and suggests an impact of aspirin on intestinal bacterial translocation to the bloodstream.

## Supporting information

**S1 File.** S1 Fig. Flow chart of data collection procedure; S1 Table. P for tests comparing relative abundances of bacterial taxa and OTUs and the odds ratios (OR) and the corresponding 95% confidence intervals (CI) of the presence of these bacteria for aspirin use. Italy 2017-2019. (DOCX)

## Acknowledgments

The authors would like to express their sincerest gratitude to all participants and collaborators to this study, without whose effort this work would not have been feasible. A special thanks to Margherita Pizzato for her essential suggestions and to Margherita Cozzi for her valuable involvement in this study. We thank Elena Tansi, Cinzia Della Noce, Rosa Restieri, Nadia Zaretti as well as all the nursing staff at the Digestive and Interventional Endoscopy Unit, ASST Grande Ospedale Metropolitano Niguarda, Milan and at the Gastroenterology and Endoscopy Unit, Fondazione IRCCS Ca' Granda Ospedale Maggiore Policlinico, Milan. The authors acknowledge support from the University of Milan through the APC initiative. Moreover, the authors acknowledge support from the Department of Clinical Sciences and Community Health, University of Milan through the APC initiative.

## Author contributions

**Conceptualization:** Silvia Mignozzi, Marcello Cintolo, Carlo La Vecchia, Marta Rossi.

**Data curation:** Marcello Cintolo, Roberto Penagini, Giorgio Gargari, Mirko Marino, Clorinda Ciafardini, Rossella Bonzi, Marta Rossi.

**Formal analysis:** Silvia Mignozzi, Marta Rossi.

**Funding acquisition:** Marta Rossi.

**Investigation:** Giuseppe De Pinto, Simone Guglielmetti, Patrizia Riso, Marcello Cintolo, Roberto Penagini, Massimiliano Mutignani, Carlo La Vecchia, Marta Rossi.

**Methodology:** Monica Ferraroni, Marta Rossi.

**Writing – original draft:** Silvia Mignozzi, Giuseppe De Pinto, Marta Rossi.

**Writing – review & editing:** Silvia Mignozzi, Giuseppe De Pinto, Simone Guglielmetti, Patrizia Riso, Marcello Cintolo, Roberto Penagini, Giorgio Gargari, Mirko Marino, Clorinda Ciafardini, Monica Ferraroni, Rossella Bonzi, Massimiliano Mutignani, Carlo La Vecchia, Marta Rossi.

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
