## [Decision Letter · Decision Letter 0]

20 Oct 2024

PONE-D-24-12563Role of aspirin on colorectal cancer risk and bacterial translocation to bloodstreamPLOS ONE

Dear Dr. Mignozzi,

Thank you for submitting your manuscript to PLOS ONE. After careful consideration, we feel that it has merit but does not fully meet PLOS ONE’s publication criteria as it currently stands. Therefore, we invite you to submit a revised version of the manuscript that addresses the points raised during the review process.

We look forward to receiving your revised manuscript.

Kind regards,

Altaf Mohammed

Academic Editor

PLOS ONE

Journal Requirements:

When submitting your revision, we need you to address these additional requirements. 1. Please ensure that your manuscript meets PLOS ONE's style requirements, including those for file naming. The PLOS ONE style templates can be found at https://journals.plos.org/plosone/s/file?id=wjVg/PLOSOne_formatting_sample_main_body.pdf and https://journals.plos.org/plosone/s/file?id=ba62/PLOSOne_formatting_sample_title_authors_affiliations.pdf 2. Thank you for stating in your Funding Statement: "Data collection was supported by the Italian Foundation for Cancer Research (AIRC) (My First AIRC grant No. 17070) (MR). Data analysis was supported by the grant PRIN 2022 PNRR (no. P20229A9S5) from the Italian Ministry of University and Research (MUR). The funders had no role in study design, data collection and analysis, decision to publish, or preparation of the manuscript." Please provide an amended statement that declares *all* the funding or sources of support (whether external or internal to your organization) received during this study, as detailed online in our guide for authors at http://journals.plos.org/plosone/s/submit-now.  Please also include the statement “There was no additional external funding received for this study.” in your updated Funding Statement. Please include your amended Funding Statement within your cover letter. We will change the online submission form on your behalf. 3. Please provide a complete Data Availability Statement in the submission form, ensuring you include all necessary access information or a reason for why you are unable to make your data freely accessible. If your research concerns only data provided within your submission, please write "All data are in the manuscript and/or supporting information files" as your Data Availability Statement. 4. We note that you have referenced ( Marino M, Mignozzi S, Michels K, Cintolo M, Penagini R, Gargari G, et al. Serum zonulin and colorectal cancer risk (Submitted). 2023.) which has currently not yet been accepted for publication. Please remove this from your References and amend this to state in the body of your manuscript: (ie “Bewick et al. [Unpublished]”) as detailed online in our guide for authorshttp://journals.plos.org/plosone/s/submission-guidelines#loc-reference-style  5. Your ethics statement should only appear in the Methods section of your manuscript. If your ethics statement is written in any section besides the Methods, please move it to the Methods section and delete it from any other section. Please ensure that your ethics statement is included in your manuscript, as the ethics statement entered into the online submission form will not be published alongside your manuscript.  6. Please include captions for your Supporting Information files at the end of your manuscript, and update any in-text citations to match accordingly. Please see our Supporting Information guidelines for more information: http://journals.plos.org/plosone/s/supporting-information.

Reviewers' comments:

Reviewer's Responses to Questions

**Comments to the Author**

1. Is the manuscript technically sound, and do the data support the conclusions?

Reviewer #1: Partly

Reviewer #2: Partly

2. Has the statistical analysis been performed appropriately and rigorously? 

Reviewer #1: No

Reviewer #2: I Don't Know

3. Have the authors made all data underlying the findings in their manuscript fully available?

Reviewer #1: Yes

Reviewer #2: Yes

4. Is the manuscript presented in an intelligible fashion and written in standard English?

Reviewer #1: Yes

Reviewer #2: Yes

5. Review Comments to the Author

Reviewer #1: The manuscript titled “Role of aspirin on colorectal cancer risk and bacterial translocation to bloodstream” explores the relationship between aspirin use, colorectal cancer, intestinal adenomas, and bacterial translocation in a case-control study. It uses both microbiome profiling and serum zonulin levels to investigate aspirin's potential role in reducing CRC and adenoma risk and its association with bacterial translocation into the bloodstream. The study provides valuable insights but has several points that require clarification before it may be suitable for publication in PLOS ONE.

The study has 100 cases for each category, which, although reasonable, results in some wide confidence intervals. This is particularly evident in the reported odds ratios (e.g., OR for CRC in aspirin users is 0.43 with a CI of 0.19–0.96). A larger sample size may help tighten these intervals and improve the study’s power to detect more robust associations. No power calculation is provided. This calculation should be included to clarify whether the study is adequately powered to detect meaningful differences between aspirin users and non-users.

Although education is adjusted for, other relevant confounders (e.g., diet, physical activity, antibiotic use, gut health history) are not mentioned or accounted for. These factors are crucial, particularly when studying gut microbiota and CRC risk. A more comprehensive adjustment for potential confounders would strengthen the findings.

There is a lack of discussion on whether blood bacterial DNA is reflective of the gut microbiome or other body sites, which could complicate interpretation of the results. While the study provides interesting microbiome data, how does this information impact current aspirin guidelines or screening for CRC? Does this suggest potential biomarkers for risk stratification? These points should be elaborated.

Minor Issues - There are minor grammatical errors throughout the manuscript.

Reviewer #2: Mignozzi et al. present an assessment of bacterial abundance in association with aspirin use among individuals with CRC and IA, compared to healthy controls. Though an interesting topic, I found the manuscript quite difficult to read and interpret. I think the manuscript requires additional detail about the procedures and execution of the study, as well as the interpretation of the data. I have listed specific concerns below, but the manuscript will require significant editing to be considered for pubication.

Line 83 – “The data was accessed for research purposes from September 2021” until when? The dates of collection and/or extraction should be included.

I feel that the methods section is significantly lacking. Further detail on the collection procedures (including consenting), sample processing, zonulin measurement are needed for reproducibility.

How was dose and duration of aspirin use accounted for in the analyses? The authors state that this information was collected in the interviews.

Why was 70 years old used as stratification in Table 1? There is clearly a difference in aspirin use by age and should be discussed.

ORs should be interpreted as reduced odds of outcome, not reduced risk.

What is the rationale for measuring serum zonulin? This is never discussed in the introduction.

Figure 1 needs to be split into multiple figures or the inlet (smaller graph) moved. The figure is very hard to interpret in its current format.

6. PLOS authors have the option to publish the peer review history of their article (what does this mean? ). If published, this will include your full peer review and any attached files.

**Do you want your identity to be public for this peer review?** For information about this choice, including consent withdrawal, please see our Privacy Policy .

Reviewer #1: No

Reviewer #2: **Yes: ** Holli Loomans-Kropp

---

## [Author Response · Author response to Decision Letter 1]

5 Dec 2024

POINT-BY-POINT RESPONSES TO REVIEWERS

Reviewer #1:

The manuscript titled “Role of aspirin on colorectal cancer risk and bacterial translocation to bloodstream” explores the relationship between aspirin use, colorectal cancer, intestinal adenomas, and bacterial translocation in a case-control study. It uses both microbiome profiling and serum zonulin levels to investigate aspirin's potential role in reducing CRC and adenoma risk and its association with bacterial translocation into the bloodstream. The study provides valuable insights but has several points that require clarification before it may be suitable for publication in PLOS ONE. The study has 100 cases for each category, which, although reasonable, results in some wide confidence intervals. This is particularly evident in the reported odds ratios (e.g., OR for CRC in aspirin users is 0.43 with a CI of 0.19–0.96).

A larger sample size may help tighten these intervals and improve the study’s power to detect more robust associations. No power calculation is provided. This calculation should be included to clarify whether the study is adequately powered to detect meaningful differences between aspirin users and non-users.

RE: We stated that “A limitation of our study is the low sample size” in the Discussion section and added the power calculation, as requested by the Reviewer: “We estimated that a total sample of approximately 200 subjects, with a ratio of controls to cases of 1, allows the assessment of an OR of 0.4 with a percentage of exposed controls of 30%, with a statistical power of 80% and a type I error of 0.05 (41). ” (line 311).

Although education is adjusted for, other relevant confounders (e.g., diet, physical activity, antibiotic use, gut health history) are not mentioned or accounted for. These factors are crucial, particularly when studying gut microbiota and CRC risk. A more comprehensive adjustment for potential confounders would strengthen the findings.

RE: We have applied a conditional logistic regression that takes into account the confounding effect of our matching variables: study center, sex, and age, besides the term of education that was included in the model. However, we agree with the Reviewer and considered other possible confounders among those suggested. We further adjusted our model for total energy intake from diet, Body Mass Index and physical activity, diabetes, and antibiotic use and obtained similar results. We modified the Methods section and added the following sentence: “When we further adjusted for BMI, physical activity, diabetes, antibiotic use, and total energy intake, the OR of CRC was 0.39 (95% CI=0.16-0.92) and the OR of IA was 0.41 (95% CI=0.18- 0.93).” (line 147) in the Results section.

There is a lack of discussion on whether blood bacterial DNA is reflective of the gut microbiome or other body sites, which could complicate interpretation of the results. While the study provides interesting microbiome data, how does this information impact current aspirin guidelines or screening for CRC? Does this suggest potential biomarkers for risk stratification? These points should be elaborated.

RE: We thank the reviewer for pointing this out. We mentioned the lack of faecal samples in the Limitations section: “Another limitation is that no other than blood samples were available. This did not allow us to evaluate the origin of the microbial materials in blood and to deepen the biochemical mechanisms of aspirin in protecting from CRC.” (line 314) In the absence of fecal samples, it is not possible to compare 16S rRNA data from blood with data from faecal and determine the origin of bacterial translocation. However, the blood samples pave the way for the possibility of metabolomic evaluations planned for future studies. We elaborated these points in the Discussion section. We added that “Our findings confirm an inverse association between aspirin use and CRC risk and suggest an impact of aspirin on bacterial translocation to explain at least in part the protective effect of aspirin against CRC. This contributes to a growing evidence on a role of aspirin use in CRC prevention, suggesting to extend in Europe the 2016 US guidelines (3) on aspirin use in CRC recommendations.” (line 322). However, our data did not allow us to define specific guidelines for subgroups of population at higher CRC risk, since there was no effect modification of aspirin use with 16S rRNA gene copies and zonulin on CRC risk.

Minor Issues - There are minor grammatical errors throughout the manuscript.

RE: We corrected typographical errors (e. g. line 233: “through” instead of “trough”; line 260: “myocardial infarction” instead of “myocardial infraction”; line 302: “correlated with” instead of “correlated to”).

Reviewer #2:

Mignozzi et al. present an assessment of bacterial abundance in association with aspirin use among individuals with CRC and IA, compared to healthy controls. Though an interesting topic, I found the manuscript quite difficult to read and interpret. I think the manuscript requires additional detail about the procedures and execution of the study, as well as the interpretation of the data. I have listed specific concerns below, but the manuscript will require significant editing to be considered for pubication.

Line 83 – “The data was accessed for research purposes from September 2021” until when? The dates of collection and/or extraction should be included.

RE: Data were collected between 1st May 2017 and 1st November 2019. We modified the following sentence in the Methods section: “We conducted a case-control study comprising 100 subjects with histologically confirmed CRC, 100 with IA and 100 patients without CRC and IA in Milan, Italy, between 1st May 2017 and 1st November 2019 (12).” (line 68). We have published the readings on the platform European Nucleotide Archive and they will be available permanently. We specified this in the Methods section: “We made all the reads publicly available in the European Nucleotide Archive (ENA) with the accession number: PRJEB46474 (uploaded on 6th September 2021 and available indefinitely). The data were used for this research purpose from September 2022 to September 2023.” (line 105). Moreover, we added further details on the data collection procedures according to the comment below.

I feel that the methods section is significantly lacking. Further detail on the collection procedures (including consenting), sample processing, zonulin measurement are needed for reproducibility.

RE: We added details on the collection procedures including exclusion criteria, recruiting procedures and additional information on the questionnaire: “Chronic bowel inflammatory diseases, liver failure, high-grade kidney and heart failure, immunodeficiency, previous cancer, hospitalization in the last month for immune, autoimmune, and inflammatory diseases were some exclusion criteria (12). Approximately 25% did not satisfy the eligibility criteria and less than 2% of eligible subjects refused to participate. Informed consent was obtained from all subjects involved in the study. Trained and blinded interviewers conducted a face-to-face interview with each subject by, before colonoscopy. The questionnaire gathered information on socio-demographics, anthropometric measures, lifestyle habits, supplement and antibiotic useand medical history. In addition, dietery information were collected through a food frequency questionnaire including 78 food items. A section included information about the use of selected drugs for more than 6 months and at least once a week, including starting date, ending date, weekly frequency and therapeutic indication. From this section, we extracted information on long-term use of aspirin 100mg/day for cardiovascular disease prevention for each subject.”

Moreover, we added more details on sample processing, 16S rRNA gene copies analysis, and zonulin measurement: “DNA extraction, quantitative polymerase chain reaction (qPCR) experiments and sequencing of 16S rRNA gene amplicons were performed by Vaiomer SAS, Labège, France (12). An aliquot of 0.25 ml of whole blood was used for DNA extraction. Real-time qPCR amplification was performed using panbacterial primers EUBF50-TCCTACGGGAGGCAGCAGT-30 and EUBR 50-GGACTACCAGGGTATCTAATCCTGTT-30. The number of 16S rRNA gene copies per µl of blood was measured as the abundance of 16S rRNA by qPCR in triplicate, normalized using a plasmid-based standard range. MiSeq Illumina technology was applied to compute 16S rRNA gene taxonomic profiling. Sequences were analyzed using Vaiomer bioinformatic pipeline, that includes the FROGS v1.4.0 to identify operational taxonomic units (OTUs) and the Blast+ v2.2.30 to perform the taxonomic assignment from Phylum to Genera, against the Silva 132 Parc database. Four samples (only one among aspirin users) were excluded from taxonomic profiling analysis because they did not reach the threshold of needed reads. We made all the reads publicly available in the European Nucleotide Archive (ENA) with the accession number: PRJEB46474 (uploaded on 6th September 2021 and available indefinitely). The data were used for this research purpose from September 2022 to September 2023. Observed-features, Chao1, Shannon, Simpson and InvSimpson alpha-diversity indices were computed using R PhyloSeq v1.14.0 package (12).

Serum zonulin levels were quantified using the ELISA kit (cat. # K5601) from Immunodiagnostik® (Bensheim, Germany) (12). Serum samples were defrosted at room temperature and applied to a 96-well plate precoated with polyclonal anti-zonulin antibodies. A biotinylated zonulin tracer was added to each sample. Peroxidase-labelled streptavidin was added to bind the biotinylated tracer. After the reaction, fluorescence was measured at 450 nm using a Tecan Infinite F200 plate reader. Zonulin levels were quantified by constructing a standard curve using a 4-parameter algorithm according to the manufacturer's instructions.”

How was dose and duration of aspirin use accounted for in the analyses? The authors state that this information was collected in the interviews.

RE: Subjects were classified as aspirin users if they reported during the interview a chronic aspirin use for cardiovascular disease prevention for more than six months. We specified in the Methods section that in the questionnaire: “A section contained information on use of selected drugs at least once a week for more than 6 months, with data on start date, end date, weekly frequency, and therapeutic indication. From this section, we extracted information on the long-term use of 100 mg/day of aspirin for the prevention of cardiovascular disease for each subject.” (line 82)

Why was 70 years old used as stratification in Table 1? There is clearly a difference in aspirin use by age and should be discussed.

RE: We have modified Table 1 according to the Reviewer suggestion, modifing to the following age categories <70, 70-74 and ≥75 . We have also included the p-value of the t-test to estimate possible differences in the mean of age between aspirin users (72.5 years) and non users (64.4 years) that was <0.0001. Moreover, also according to the Reviewer 1 comment, we specified in the Methods that the relatioship between aspirin use and CRC risk were estimated using logistic model conditioned on the matching variables, that were study center, sex, age (±5). Therefore, we took into account the possible confounding effect of age in our analysis. We added in the Discussion section: “With reference to confounding, we were able to allow for a number of relevant covariates, including study center, sex, and age.” (line 309).

Aspirin use Non aspirin use Χ2 p-value

n=58 n=242

n. (%) n. (%)

Age group

<70 19 (32.8) 155 (64.1)

70-74 14 (24.1) 34 (14.1)

≥75 25 (43.1) 53 (21.9)

<0.0001

Mean (SD) 72.5 (8.0) 64.4 (11.6) <0.0001b

ORs should be interpreted as reduced odds of outcome, not reduced risk.

RE: The odds ratio, which represents the odds of developing the disease in exposed subjects compared with the odds in non-exposed subjects, can be approximately equal to the risk ratio if the risk of the disease is low in both exposed and non-exposed groups. This condition is met to a reasonable approximation when the disease is rare, as is CRC. However, the odds ratio (introduced by Cornfield in his JNCI 1951;11:1269-75 publication) has been shown by Miettinen (Am J Epidemiol. 1976;103:226-35) to be an unbiased estimation of the relative risk (more specifically, the incidence rate ratio) in case-control studies based on incident cases, as in our study, and this is also discussed in standard textbooks (see, for instance: MacMahon B and Trichopoulos D. Epidemiology: Principles and Methods – 2nd Ed. Little Brown, Boston, USA 1996). The relative risk can and has been calculated in thousands case-control studies through the odds ratio. In short, the odds ratio is the most commonly used relative risk estimate in case-control studies.

What is the rationale for measuring serum zonulin? This is never discussed in the introduction.

RE: Zonulin is the only known human protein able to of regulate intestinal permeability by modulating intercellular tight junctions. It controls the opening and closing of these junctions, allowing molecules and fluids to pass through. Zonulin plays a key role in protecting the intestinal barrier against bacterial translocation. We included the following sentence in the Introduction: “In addition circulating levels of zonulin -the protein responsible for regulating tight junctions and crucial for protecting the intestinal barrier against bacterial translocation-and blood bacterial 16S rRNA gene copies-a measure of blood bacterial load- have been positively associated with CRC risk (11-14).” (line 58).

Figure 1 needs to be split into multiple figures or the inlet (smaller graph) moved. The figure is very hard to interpret in its current format.

RE: According to the Reviewer suggestion, we splitted Figure 1 into Figures 1a-1g.

---

## [Decision Letter · Decision Letter 1]

30 Dec 2024

PONE-D-24-12563R1Role of aspirin on colorectal cancer risk and bacterial translocation to bloodstreamPLOS ONE

Dear Dr. Mignozzi,

Thank you for submitting your manuscript to PLOS ONE. After careful consideration, we feel that it has merit but does not fully meet PLOS ONE’s publication criteria as it currently stands. Therefore, we invite you to submit a revised version of the manuscript that addresses the points raised during the review process.

Please carefully address reviewer's comments.

We look forward to receiving your revised manuscript.

Kind regards,

Altaf Mohammed

Academic Editor

PLOS ONE

Additional Editor Comments:

Please carefully address reviewer's comments.

Reviewers' comments:

Reviewer's Responses to Questions

**Comments to the Author**

1. If the authors have adequately addressed your comments raised in a previous round of review and you feel that this manuscript is now acceptable for publication, you may indicate that here to bypass the “Comments to the Author” section, enter your conflict of interest statement in the “Confidential to Editor” section, and submit your "Accept" recommendation.

Reviewer #1: (No Response)

Reviewer #2: (No Response)

2. Is the manuscript technically sound, and do the data support the conclusions?

Reviewer #1: (No Response)

Reviewer #2: Partly

3. Has the statistical analysis been performed appropriately and rigorously? 

Reviewer #1: (No Response)

Reviewer #2: No

4. Have the authors made all data underlying the findings in their manuscript fully available?

Reviewer #1: (No Response)

Reviewer #2: Yes

5. Is the manuscript presented in an intelligible fashion and written in standard English?

Reviewer #1: (No Response)

Reviewer #2: No

6. Review Comments to the Author

Reviewer #1: (No Response)

Reviewer #2: Overall comment: Still needs to be edited for grammatical issues (e.g., comma placement, colons used instead of semicolons, etc) and sentence structure. It is still difficult to read. The results should be written to actually present the results of the study, rather than just stating what the figures and tables show. Figures/tables should be referenced in the text and not be the focus.

Aspirin use is no longer recommended by the USPSTF for CRC prevention (lines 45-46 and 321-322; please see doi:10.1001/jama.2022.4983). The introduction needs to be adjusted to reflect the updated guidance.

A study flow chart would be appropriate to add as figure 1 to clearly outline participant eligibility criteria and final study population.

For clarification purposes, was an ‘aspirin user’ one who used aspirin at 100 mg/day for at least 6 months?

Medical history was collected in the questionnaire. Because aspirin is often taken for MI and other cardiac conditions, is there a reason why history of MI or CVD is not controlled for in the model?

Results – Please add percentages for education (line 140).

Figure 1 legend – Each subpart of the figure doesn’t need its own legend. Please combine into a single figure legend.

Lines 312-315 – I think the authors just need to state that the study was underpowered, rather than provide the power calculation.

7. PLOS authors have the option to publish the peer review history of their article (what does this mean? ). If published, this will include your full peer review and any attached files.

**Do you want your identity to be public for this peer review?** For information about this choice, including consent withdrawal, please see our Privacy Policy .

Reviewer #1: No

Reviewer #2: **Yes: ** Holli Loomans-Kropp

---

## [Author Response · Author response to Decision Letter 2]

29 Jan 2025

Editor-in-Chief:

PLOS ONE

Prof. Emily Chenette, PhD

Milan, January 27th 2025

Dear Prof. Chenette,

Thank you for considering our manuscript entitled “Role of aspirin on colorectal cancer risk and bacterial translocation to bloodstream”. Please, find enclosed a revised version according to the suggestions of Reviewers. Hereafter, we give the point-by-point answers the Reviewer comments.

Thank you in advance for your kind attention.

Yours Sincerely,

Dr. Silvia Mignozzi

Department of Clinical Sciences and Community Health

Branch of Medical Statistics, Biometry and Epidemiology "G.A. Maccacaro"

Università degli Studi di Milano

Via G. Celoria 22, 20133 Milan, Italy

Email: silvia.mignozzi@unimi.it;  

POINT-BY-POINT RESPONSES TO REVIEWERS

Reviewer #2:

Overall comment: Still needs to be edited for grammatical issues (e.g., comma placement, colons used instead of semicolons, etc) and sentence structure. It is still difficult to read.

We edited and revised the manuscript, modifying the placement of commas and colons. We restructured some sentences, especially in the Results and Discussion.

The results should be written to actually present the results of the study, rather than just stating what the figures and tables show. Figures/tables should be referenced in the text and not be the focus.

We modified the Results as suggested by the Reviewer (lines 135-171 ). In particular we changed: “Aspirin users had higher levels of serum zonulin (median =30.7 ng/mL ) as compared to non-users (median=28.5 ng/mL) among controls (p=0.02) (Table 3). Aspirin users had lower levels of serum zonulin (median=27.2) as compared to non-users (median=29.1) among IA cases, without a significant difference (p=0.1). No difference was found in terms of zonulin, in CRC cases and overall. With reference to 16S rRNA gene analysis, higher copies were found in aspirin users (median=7786.2) versus non-users (median=7084.5) overall, without a significant difference (p=0.1). Similarly, higher copies were found in aspirin users (median=7794.8 copies per μL) versus non-users (median=6970.2 copies per μL) among controls (p=0.07). We found a lower bacterial richness in aspirin users compared to non-users overall, in genera in terms of both Observed-features (median=27.0 vs. 29.0, p=0.02) and Chao1 (median=37.0 vs. 45.0, p=0.004) indices and in OTUs in terms of both Observed-features (median=34.0 vs. 36.0, p=0.02) and Chao1 (median=54.5 vs. 66.0, p=0.04) indices overall. Similar results were found among CRC cases and controls. No difference between aspirin users and non-users was found in terms of Shannon, Simpson and InvSimpson alpha-diversity indices.”

Aspirin use is no longer recommended by the USPSTF for CRC prevention (lines 45-46 and 321-322; please see doi:10.1001/jama.2022.4983). The introduction needs to be adjusted to reflect the updated guidance.

We thank the Reviewer for the comment and adjusted the manuscript to reflect the updated literature, adding the reference doi:10.1001/jama.2022.4983, doi:10.1001/jama.2022.3337, 10.1002/ijc.35331. We made the Introduction and the Discussion more cautious, as suggested.

We modified the last sentence as follows: “This study provides additional data on the possible inverse association between aspirin use and CRC risk (42, 43) and suggests an impact of aspirin on intestinal bacterial translocation to the bloodstream.”

A study flow chart would be appropriate to add as figure 1 to clearly outline participant eligibility criteria and final study population.

As suggested by the Reviewer, we added a Supplementary Figure 1 showing the data collection procedure.

For clarification purposes, was an ‘aspirin user’ one who used aspirin at 100 mg/day for at least 6 months?

Yes, in our study an ‘aspirin user’ is defined as a subject who reported aspirin use at ≥100 mg/day for at least 6 months. As suggested by the Reviewer, we specified in the Methods “From this section, we extracted information to identify a long-term aspirin user as a subject who reported aspirin use at ≥100 mg/day for at least 6 months for cardiovascular prevention.”

Medical history was collected in the questionnaire. Because aspirin is often taken for MI and other cardiac conditions, is there a reason why history of MI or CVD is not controlled for in the model?

According to suggestion of the Reviewer, we further adjusted the the model for cardiovascular diseases and obtained similar results. We modified the sentence in the Results showing the ORs from a fully adjusted model including a term for cardiovascular diseases: “The OR of CRC was 0.38 (95% CI=0.16-0.91) and the OR of IA was 0.41 (95% CI=0.18-0.94) from the fully adjusted model.”

Results – Please add percentages for education (line 140).

We added percentages as suggested by the Reviewer.

Figure 1 legend – Each subpart of the figure doesn’t need its own legend. Please combine into a single figure legend.

We modified the figure legend as suggested by the Reviewer.

Lines 312-315 – I think the authors just need to state that the study was underpowered, rather than provide the power calculation.

We specified that the study was underpowered as suggested by the Reviewer (line 287).

---

## [Editor Report · Decision Letter 2]

7 Feb 2025

Role of aspirin on colorectal cancer risk and bacterial translocation to bloodstream

PONE-D-24-12563R2

Dear Dr. Mignozzi,

We’re pleased to inform you that your manuscript has been judged scientifically suitable for publication and will be formally accepted for publication once it meets all outstanding technical requirements.

Kind regards,

Academic Editor

PLOS ONE

---

## [Editor Report · Acceptance letter]

PONE-D-24-12563R2

PLOS ONE

Dear Dr. Mignozzi,

I'm pleased to inform you that your manuscript has been deemed suitable for publication in PLOS ONE. Congratulations! Your manuscript is now being handed over to our production team.

Kind regards,

on behalf of

Dr. Altaf Mohammed

Academic Editor

PLOS ONE